# Effect of KCl Addition on First Hydrogenation Kinetics of TiFe

**Joydev Manna [1] and Jacques Huot [2],* [ID]**

[1]   Hydrogen Energy Division, National Institute of Solar Energy, Gurugram 122003, India
[2]   Institut de Recherche sur L'hydrogène, Université du Québec à Trois-Rivières, 3351 des Forges, Trois-Rivieres, QC G9A 5H7, Canada
*   Correspondence: jacques.huot@irh.ca; Tel.: +1-819-376-5011

**Abstract:** In this paper, the effect of the addition of potassium chloride (KCl) by ball milling on the first hydrogenation kinetics of TiFe is reported. After milling, KCl was uniformly distributed on the TiFe's surface. As-synthesized TiFe does not absorb hydrogen. However, after ball milling with KCl, it absorbed 1.5 wt.% of hydrogen on the first hydrogenation without any thermal treatment. The storage capacity of TiFe with KCl addition is higher than that of the ball milled pure TiFe. The effects of the amount of KCl additive in TiFe and ball milling time on first hydrogenation kinetics are reported. It is noted that, with an increase in KCl amount and ball milling time, hydrogenation kinetics are improved. However, hydrogen storage capacity decreased for both cases.

**Keywords:** hydrogen storage; TiFe; KCl; ball milling; hydrogenation; kinetics

## 1. Introduction

Intermetallic hydrides (IMHs), especially $LaNi_5$ and TiFe, are of great interest due to their reversible hydrogen absorption-desorption abilities [1]. In particular, titanium–iron (TiFe) intermetallic compound (IMC) is considered promising for hydrogen storage applications due to its abundance, low pyrophoricity, low costs and adequate reversible hydrogen storage capacity (~1.9 wt.%) at low pressures (1–2 MPa) and temperatures (30–70 °C) [2–6]. TiFe's crystal structure is CsCl-type and shows very fast reaction kinetics during hydrogen absorption–desorption cycles [7]. During the hydrogenation of TiFe, three phases, namely, TiFeH solid solution (α), FeTiH monohydride (β) and $FeTiH_2$ dihydride (γ), are formed [8].

However, TiFe IMC and its derivatives suffer from poor first hydrogenation (also called activation) performances and low poisoning tolerance in the presence of trace amounts of oxidative gases such as oxygen and water vapor. The first hydrogenation of TiFe samples prepared by conventional methods (e.g., arc-melting, induction melting, etc.) is a difficult and energy-intensive process [9–11]. This happens mainly due to the formation of a surface oxide layer of $TiO_2$ and/or $Fe_2O_3$ during the synthesis process or air exposure of TiFe [12]. The surface oxide layer blocks the metal–hydrogen electron interactions and prevents the hydrogenation process. Mechanical processes such as ball milling, Equal Chanel Angular Pressing (ECAP), High-Pressure Torsion (HPT), and cold rolling (CR) are found to be effective on the activation of TiFe [13–19]. It is suggested that these processes can introduce non-equilibrium phase, a nanoscale structure, and active sites such as defects or grain boundaries that can facilitate the hydrogenation kinetics.

On the other hand, partial substitution of the main components with other elements have also been considered as an alternative approach to improve the hydrogen storage properties of intermetallic compounds [20,21]. Presence of a third component such as Zr, V, Cr, Mn, Co, Ce, Nb or Y in TiFe could improve the first hydrogenation kinetics [22–28]. The effect of other additives such as $Zr_7Ni_{10}$, $ZrMn_2$, etc., on the activation of TiFe was also studied [29,30]. Room temperature oxidation of Mn-doped TiFe alloys by $O_2$ and $H_2O$ was studied by Shwartz et al. and they reported that TiO and $TiO_2$ formed after exposure [31].

It has also been reported that LaNi$_5$ has better cyclic stability under impure H$_2$ gas than TiFe [32]. Reactivation of poisoned TiFe-based alloys was studied by our group [9,12,17,33] as well as by Modi and Aguey-Zinsou [34]. An addition of a small amount of Mn and Ce to TiFe is also found to be effective towards activation processes [27]. Qu et al. observed that the addition of cobalt (Co) can make the activation process easier by flattening the hydrogen absorption and desorption plateau [35]. Gosselin et al. reported that a small amount of Y can simplify the activation procedure of TiFe [36]. Ali et al. has tested the effect of addition of Y and Cu to the TiFe$_{0.86}$Mn$_{0.1}$ and observed formation of a secondary phase (Cu$_4$Y), which accelerates the first hydrogenation process [23,24]. Jain et al. has observed a positive effect of Zr$_7$Ni$_{10}$ addition to TiFe on its activation process [30]. In most of these cases, it is reported that secondary phase formation accelerates the first hydrogenation kinetics. Detailed reviews of the microstructure, hydrogenation kinetics and methods for improvement of the activation process of TiFe IMC and its derivatives are also reported recently [6,37].

It has also been reported that surface modification by means of nano layer formation [38–41] and Pd or Ni deposition [42–49] are effective towards activation of TiFe. It has been found that TiFe + Pd thin films with a Pd coating of about 20 nm on 100–200 nm TiFe facilitates the first hydrogenation of TiFe and kinetics can be improved by annealing of the samples [49]. Similarly, a small amount of pure Ni doping on TiFe surface readily absorbed around 0.6 wt.% of hydrogen without any activation process [43]. However, for practical applications, the use of precious metals for activation of TiFe will not be cost-effective. Therefore, in this paper, the effect of the addition of potassium chloride (KCl) salt on first hydrogenation kinetics of TiFe was studied, mainly due to the high abundance, and low cost of KCl compared to the Pd or Ni metals. A significant improvement in the hydrogenation kinetics has been noticed for the KCl-modified TiFe compared to the pure TiFe. The effects of salt amount and mechanical milling time on activation kinetics were studied. Cyclic stability as well as the structure, morphology and chemical composition of the samples were investigated. The effects of other salts (NaCl, CaCl$_2$, KI, and FeCl$_3 \cdot 6H_2O$) are also reported.

## 2. Materials and Methods

TiFe was synthesized by arc-melting processes using industrial grade Fe (ASTM 10005) and Ti (ASTM B265 grade 1) metals. Arc melting was performed under an Ar atmosphere at 240 V and 60 A. The pellet obtained during arc melting was turned over three times and remelted to ensure homogeneity.

Ball milling (BM) was performed using a SPEX 8000M high energy ball mill machine (Spex® SamplePrep, Metuchen, NJ, USA). Before milling, an as-cast TiFe pellet was crushed and properly mixed with 1 wt.% of KCl (99%, Alfa-Aesar (Tewksbury, MA, USA)) salt using a mortar and pestle under Ar atmosphere. The obtained mixture was then taken inside a hardened steel crucible (volume 55 cc) with a powder to ball ratio of 1/10 and milled for 30 min. The salt amount (1 and 5 wt.%) and ball milling time (10, 30 and 60 min) were varied separately to observe the effect of these two important parameters on the first hydrogenation kinetics. All samples were handled inside an argon-filled glove box. To observe the effect of other salts on the activation of TiFe, a set of samples were prepared by milling as-crushed TiFe and 1 wt.% of each of the salts (sodium chloride (NaCl), calcium chloride (CaCl$_2$), potassium iodide (KI), iron chloride (FeCl$_3 \cdot 6H_2O$)), separately, for 30 min following a similar method as described above.

A homemade Sieverts apparatus was used for the hydrogenation experimentation. Hydrogenation experiments were performed at room temperature (298 K) under 20 bar hydrogen pressure (ultra-high purity H$_2 \geq$ 99.999%). Samples were kept under dynamic vacuum at room temperature for 1 h before each measurement. No heat treatment was given to any of the samples before or during the hydrogenation experimentation.

Crystal structure and phase composition were determined by X-ray diffraction (XRD) using a Bruker D8 Focus (Bruker, Madison, WI, USA) with Cu K$\alpha$ radiation. Rietveld's

method, using TOPAS software (Version 6, Bruker, Madison, WI, USA) was used to evaluate the lattice parameters of materials [50]. Microstructure and chemical analysis were performed using a JEOL JSM-5500 scanning electron microscope (JEOL, Peabody, MA, USA) equipped with an EDX (Energy Dispersive X-ray) apparatus from Oxford Instruments (Abingdon, UK).

## 3. Results and Discussion

### 3.1. Characterization

Crystal structures and phase compositions of TiFe in as-cast samples and those milled with KCl were analyzed using powder X-ray diffraction (XRD). XRD patterns of as-cast TiFe, TiFe + 1 wt.% KCl (30 min BM) and KCl salt are shown in Figure 1a–c. The pattern of as-cast TiFe shows Bragg's peaks associated with TiFe's CsCl structure (Pm-3m space group) with a lattice parameter $a$ = 2.9805 Å (PDF#65-5613). Two small peaks around 41.5° and 45.5° can also be observed, corresponding to a secondary $Fe_2Ti$ phase (MgZn$_2$-type, P63/mmc space group) in the as-cast sample. As shown in Figure 1b, the diffraction pattern of pure KCl salt is indexed to the FCC structure of KCl (PDF# 41-1476).

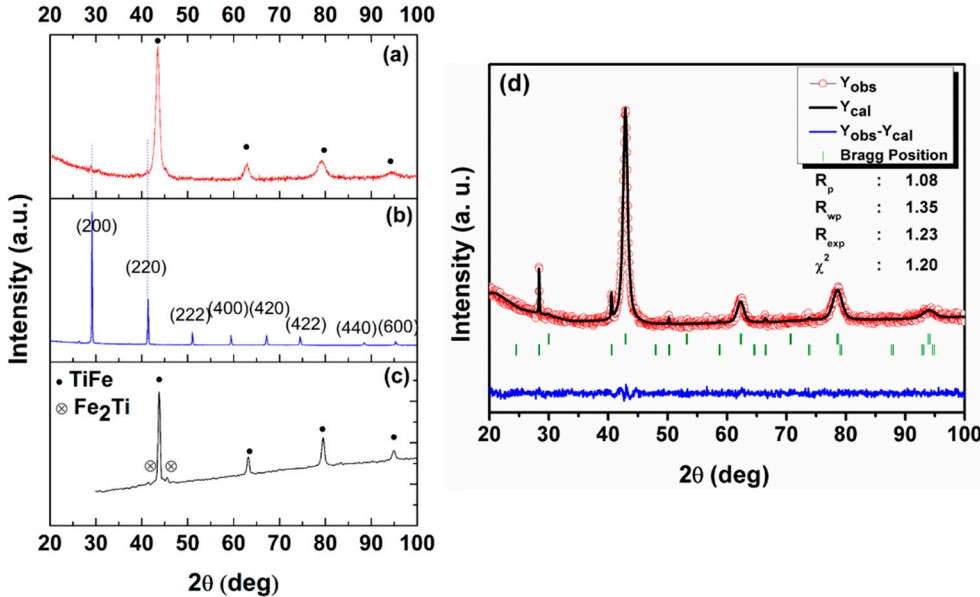

**Figure 1.** X-ray diffraction patterns of (**a**) TiFe + 1 wt.% KCl 30 min BM; (**b**) KCl; (**c**) TiFe; (**d**) TiFe + 5 wt.% KCl 30 min BM with Rietveld refinement. In (**d**), the upper tick mark indicates the Bragg peaks of the TiFe phase and the lower tick marks the KCl phase.

Phase composition, lattice parameters and crystallite size of the as-cast samples and KCl-modified TiFe samples were determined using the Rietveld refinement method (Figure 1d) and shown in Table 1. In the diffraction pattern of TiFe with KCl addition, the KCl abundance derived from Rietveld's analysis agrees with the nominal values. Interestingly, no substantial amount of the $Fe_2Ti$ phase was observed after milling with KCl. This may be due to the reduction of crystallite size, which broadens the Bragg peaks and makes them difficult to distinguish from the background. As expected, ball milling significantly reduces the crystallite size and particle size of the materials. The crystallite size of TiFe phase decreased from 23.8 nm to ~9 nm. The ball milling process also increased the microstrain of the TiFe phase. Because of their low abundances, the microstrain of $Ti_2Fe$ and KCl could not be refined. The metastable $Ti_2Fe$ phase has not been studied for its hydrogenation properties due to its difficult synthesis process [17]. The lattice parameters of TiFe and its derivative IMCs vary from a lower boundary of 2.953(2) Å to an upper boundary of 2.9802(2) Å [51,52]. The lattice parameter obtained for TiFe and KCl-modified TiFe agrees with the reported value [33,53–57].

**Table 1.** Phase composition, lattice parameters, crystallite size and microstrains of TiFe, and KCL-modified TiFe. Numbers in parentheses are uncertainties for the last significant digit.

| Sample | Phase | Composition (wt.%) | Lattice Parameter (Å) | Crystallite Size (nm) | Microstrain (%) |
|---|---|---|---|---|---|
| As-cast TiFe | TiFe | 95(1) | 2.9808(3) | 23.8(7) | 0.092(4) |
| | $Fe_2Ti$ | 5(1) | $a = 4.840(4), c = 8.11(1)$ | 32(12) | – |
| TiFe + 1 wt.% KCl | TiFe | 98.6(3) | 2.9815(6) | 8.5(2) | 0.269(7) |
| | KCl | 1.4(3) | 6.2849 | 50(20) | – |
| TiFe + 5 wt.% KCl | TiFe | 95.4(3) | 2.9806(2) | 9.1(2) | 0.264(7) |
| | KCl | 4.6(3) | 6.2849 | 89(13) | – |

The SEM micrographs of crushed samples of as-cast TiFe and ball-milled TiFe (without any additives) are shown in Figure 2a,b. The TiFe after crushing with a mortar and pestle formed relatively large particles of uneven shapes with sharp edges. The particle size distribution is from 90 μm to 500 μm. An unambiguous difference can be seen in the samples before and after milling. After ball milling of pure TiFe, a mix of particles with small particles of a few nanometers along with a few larger particles having sizes in the range of few hundreds of micrometers was seen. The particle size of the sample after milling was found to be in the range of about 50 nm–400 μm. Surfaces of the larger particles were observed to be covered with smaller, aggregated, round-shaped particles.

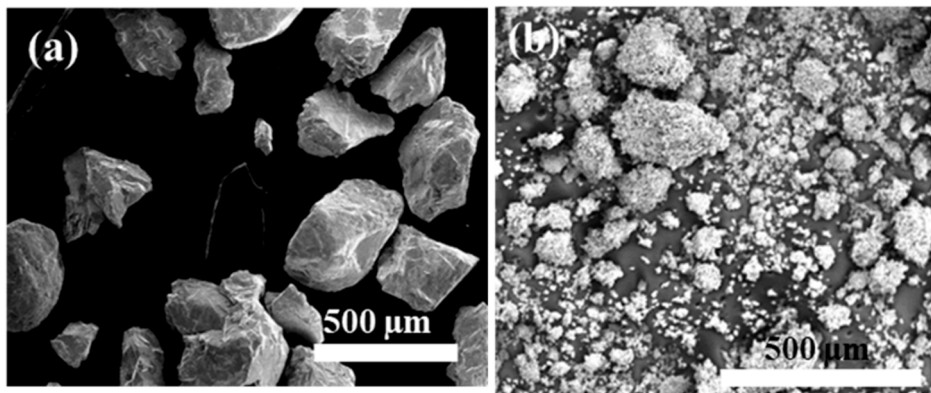

**Figure 2.** SEM images of (**a**) as-crushed TiFe and (**b**) 30 min BM TiFe samples.

Figure 3 shows SEM images and particle size distribution of the TiFe + 1 wt.% KCl sample after ball milling for different time intervals (10, 30 and 60 min). As it can be seen from the figure, particle size decreases with milling time. It can be noted from the figure that for the 10 min milled sample, most of the particles were found to be in the range of 2–80 μm with some particles having a larger size (100–150 μm). However, larger particles (>50 μm) were not present after milling for 30 and 60 min. From the particle size distribution curve (Figure 3d–f), it can be noted that the average particle size of the KCl addition samples after ball milling for 30 and 60 min was 9.56(3) and 8.66(2) μm, respectively. The milled samples also have many particles in the size of nanometers and the quantities of these particles increased with milling time. It has also been noted that after milling for a longer time (60 min) there was a tendency of samples to adhere on the surface of the milling bowl. This may lead to a low yield of sample recovery after ball milling for a long time. This adherence can be avoided by using a liquid milling agent such as a solvent, as we have performed for pure TiFe in our previous study [9]. Several works already reported that the plastic deformation done by milling, high-pressure torsion or their combination may reduce the particle and crystallite sizes of the TiFe IMCs as well as generates grain boundaries and micro-structural defects. Similarly, it has also been reported that these processes may create an amorphous phase [12].

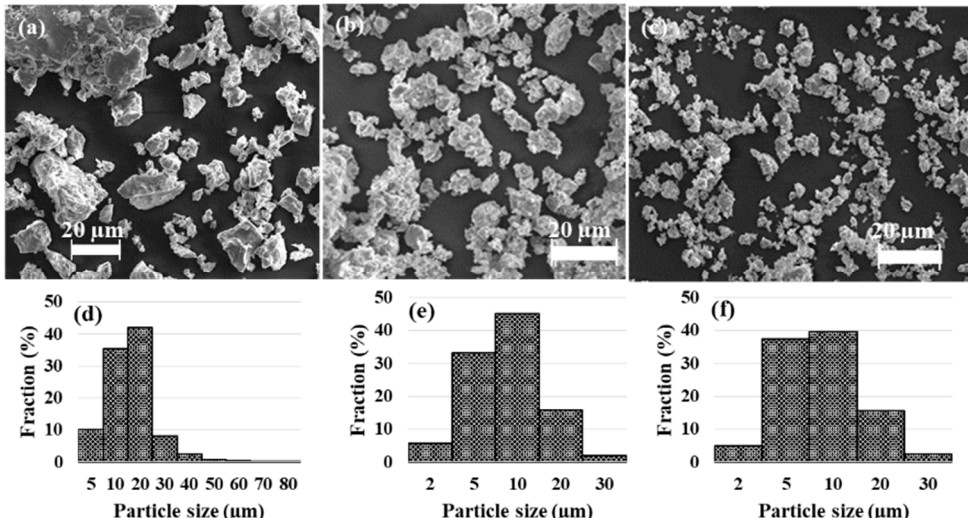

**Figure 3.** (**a–c**) SEM images of TiFe + 1 wt.% KCl samples after milling for 10 min, 30 min and 60 min, respectively and (**d–f**) particle size distribution of TiFe + 1 wt.% KCl sample after milling for 10, 30 and 60 min, respectively.

The EDX analysis of the KCl-modified sample (TiFe + 1 wt.% KCl, 30 min BM) is shown in Figure 4. As can be seen from the Figure, the EDX analysis indicates that KCl is almost exclusively distributed throughout the surface of the TiFe particles. No aggregation of KCl or cluster formation can be seen. Around 1.5 at% of potassium (K) and 1.4 at% of chlorine (Cl) could be found from the EDX measurements on the surface of the TiFe + 1 wt.% KCl sample.

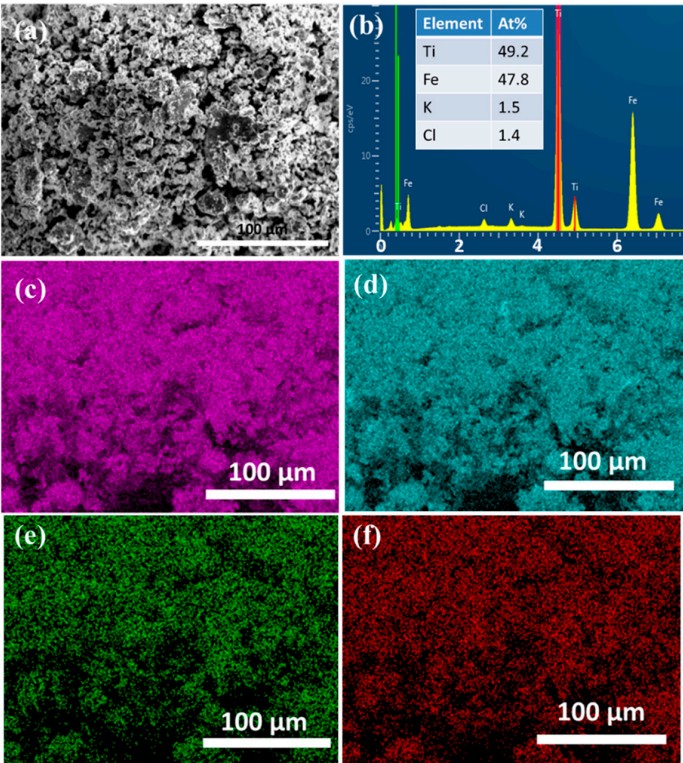

**Figure 4.** EDX analysis of TiFe + 1 wt.% KCl 30 min BM sample. (**a**) SEM micrograph, (**b**) elemental peak profiles by EDX with elemental composition (in at%) shown in table, (**c**) Ti (**d**) Fe (**e**) K (**f**) Cl elemental mapping.

### 3.2. Activation Kinetics

The first hydrogenation kinetic curves of the as-cast TiFe, ball-milled (30 min) TiFe and TiFe + 1 wt.% KCl (BM for 30 min) samples are shown in Figure 5. The samples were kept under a dynamic vacuum at room temperature for 1 h prior to each hydrogenation experiments. As can be seen from the Figure 5, the as-cast TiFe did not absorb any hydrogen during the first hydrogenation. Ball milling of the pure TiFe for 30 min causes a significant improvement in first hydrogenation kinetics, attaining a storage capacity of ~1.1 wt.% on the first cycle without any incubation time. Thus, it could be noted that ball milling of the pure TiFe can facilitate the first hydrogenation without any other pre-treatment. A similar observation has been reported for BM TiFe IMCs by several groups [58–60]. The ball milling processes reduces the particle and crystallite sizes while increasing the microstrain. A smaller crystallite size means that the amount of grain boundaries is higher which helps the hydrogen diffuse in the alloy. These factors contribute greatly to improve the first hydrogenation kinetics after ball milling [33,61]. The inertness of the as-cast TiFe is mainly due to the formation of oxide layers ($TiO_2$, $Fe_2O_3$, FeO) on the alloy's surface [8]. After ball milling, hydrogenation kinetics were improved due to the formation of smaller particles, reduction of crystallite size and the formation of grain boundaries as suggested by previous studies [58,61]. The reduction of hydrogen storage capacity after milling is probably due to the formation of amorphous phases such as grain boundaries [33].

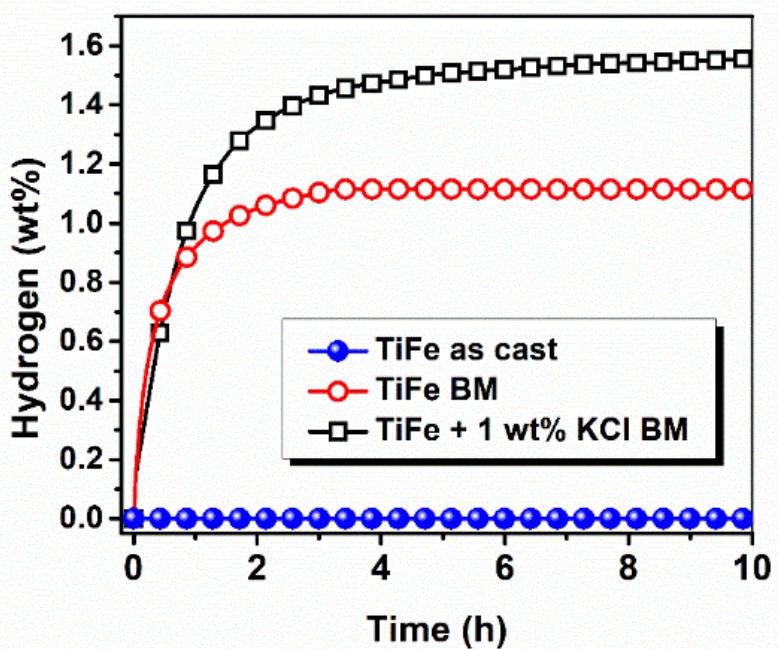

**Figure 5.** First hydrogenation kinetics at 298 K under 20 bar $H_2$ pressure of as-cast TiFe, ball-milled TiFe and ball-milled TiFe + 1 wt.% KCl.

The TiFe + 1 wt.% KCl 30 min BM sample presents similar initial kinetics as those of the pure TiFe 30 min BM sample, but it absorbed more hydrogen (Figure 5). A maximum of 1.56 wt.% storage capacity was noted for this sample after 10 h of hydrogen exposure. The increase in storage capacity for KCl-modified samples is not clear and may need further investigation. However, it has been noted that the storage capacities of both the ball-milled samples (as-cast and KCl-modified) were found to be less compared to the nominal storage capacity of TiFe (1.86 wt.%, [62]). The decrease in storage capacity is mostly due to the formation of grain boundaries after the milling process and has been noted in the case of several TiFe-based alloys [63–67]. The formed grain boundaries can easily enhance the first hydrogenation kinetics while acting as a diffusion route for $H_2$. However, they are not able to form M–H bonds. This results in reduction of total capacity. A comparison of

activation of TiFe IMCs are shown in Table 2. As can be seen from the table, a wide range of methods have been employed for the activation of TiFe IMCs and hydrogen storage capacity of pure TiFe has been found to be in the range of 1.1–1.5 wt.% depending upon the activation methods and reaction condition during the first hydrogenation process.

**Table 2.** Comparison of activation processes of TiFe IMCs.

| Sample | Synthesis Process | Air Exposure | Treatment | | $H_2$ Adsorbed | Ref |
| | | | Mechanical | Thermal | | |
|---|---|---|---|---|---|---|
| TiFe | Induction melting | As-cast | Ball milling | No | 1.1 wt.% (20 bars, 20 °C) | [12] |
| TiFe | Mechanical alloying | - | - | 5 cycles of vacuum and 150 bar $H_2$ pressure at 300 °C | 1.3 wt.% | [65] |
| TiFe | Purchased | - | Ball milling | Annealing (1000 °C) | 1.3–1.5 wt.% (100 bar, 30 °C) | [58] |
| TiFe | Purchased | | Ball milled with ethanol | - | 1.2 wt.% (RT) | [68] |
| TiFe | Arc melting | | Ball milling with 1% KCl | - | 1.5 wt.% (RT, 20 bars) | This study |
| TiFe$_{0.85}$Mn$_{0.15}$ | Arc-melting | 2 h | Ball milling | 300 °C under 3 MPa $H_2$ pressure, 3 times | <1 wt.% (30 bars, 30 °C) | [34] |
| TiFe + 2 wt.% Mn + 4 wt.% Zr | Gas atomization | 60 days | Cold rolling | No | 2.1 wt.% (20 bars, RT) | [17] |

To further optimize the KCl amount, another TiFe + KCl sample was prepared via milling for 30 min with 5 wt.% of KCl. Figure 6 shows the typical hydrogen absorption curves for different amounts of KCl addition to TiFe. As it can be seen from Figure 6, the initial hydrogenation kinetics rate is faster for the 5 wt.% KCl-modified sample compared to the 1 wt.% KCL-modified sample. However, the total storage capacity of 5% KCL-modified TiFe is found to be around 1.34 wt.% which is less than the 1 wt.% KCL-modified sample. Evidently, KCl could not absorb hydrogen by itself. However, adding 1 wt.% or 5 wt.% of KCl reduced the possible maximum capacity from 1.86 wt.% to respectively 1.84 wt.% and 1.77 wt.%. This means that another cause for the reduction of capacity is in play. As seen before, milling with KCl makes the crystallite size smaller and increases the amount of grain boundaries. The grain boundaries are gateways for hydrogen and thus makes the kinetics faster, but they do not store hydrogen. This may be the reason for the decrease in hydrogen storage capacity with a higher salt amount, but further investigation is needed to clarify this point.

The effect of ball milling time on the first hydrogenation kinetics was investigated for the TiFe + 1 wt.% KCl sample and the results are shown in Figure 7. It can be seen that the hydrogenation kinetics improve with milling time. For the 10 min milled sample, a short incubation time was noted but it took more than 10 h to reach about 1.1 wt.% hydrogen capacity. For the 1 h milled sample, about 1.4 wt.% of hydrogen was absorbed within 2 h of hydrogen exposure. With increasing milling time, the hydrogen storage capacity of the samples decreases due to the formation of more defects, grain boundaries, etc.

A set of cycling tests were performed on the TiFe + 1 wt.% KCl 30 min BM sample. The hydrogenation kinetics after four cycles are shown in Figure 8. It can be seen that the hydrogenation kinetics decrease to around 0.9 wt.% after the first hydrogenation. Cycling of metal hydrides causes mechanical stress in the crystals and it is known that MH expands during hydrogenation and contracts during desorption, resulting in the pulverization of metal particles [24]. Thus, loss in hydrogen storage capacity could be seen in most of the metal alloys or intermetallic compounds when cycled. In the present case, the storage capacity of around 0.9 wt.% was found to be constant up to the fourth absorption. The reason for this loss of capacity is unclear and the investigation is underway on this matter.

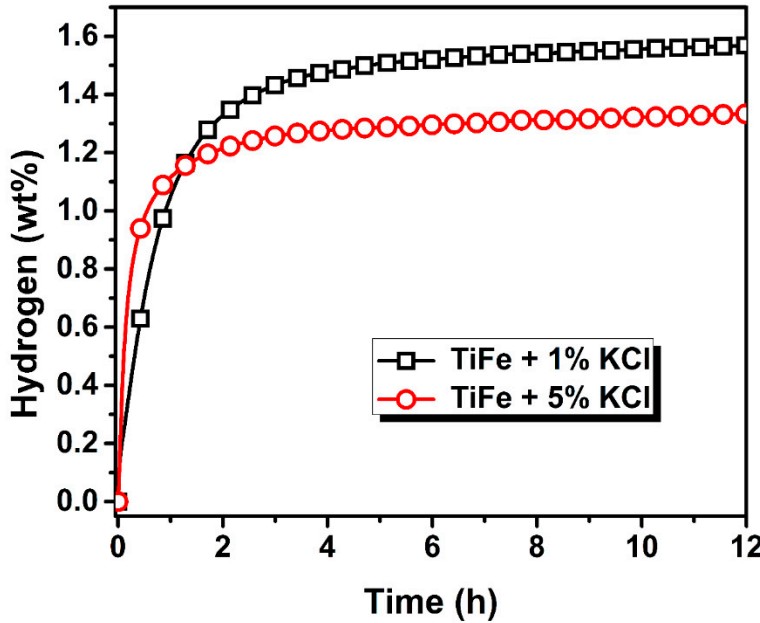

**Figure 6.** Effect of amount of KCl additives on first hydrogenation kinetics of TiFe at 298 K under 20 bars of hydrogen pressure.

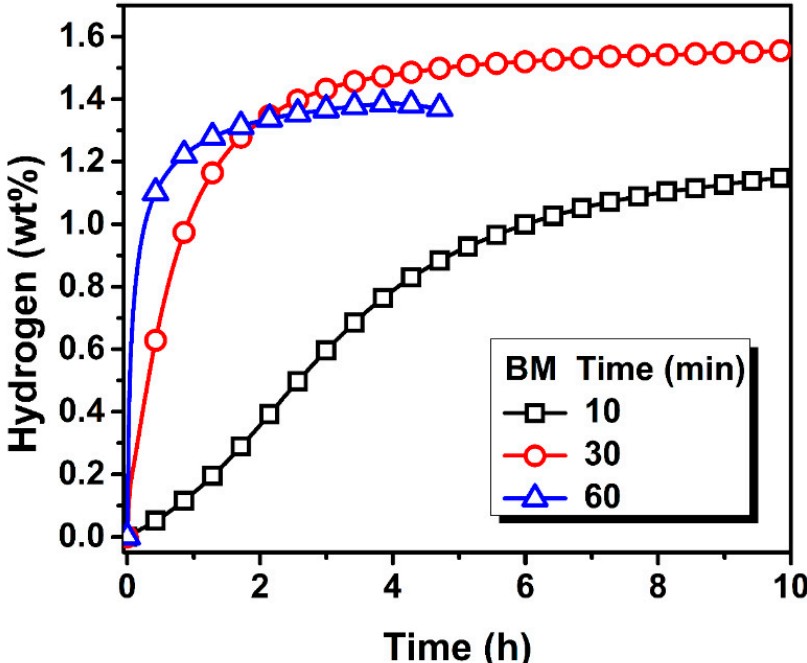

**Figure 7.** Effect of ball milling time on the first hydrogenation kinetics of TiFe + 1 wt.% KCl at 298 K under 20 bars of hydrogen pressure.

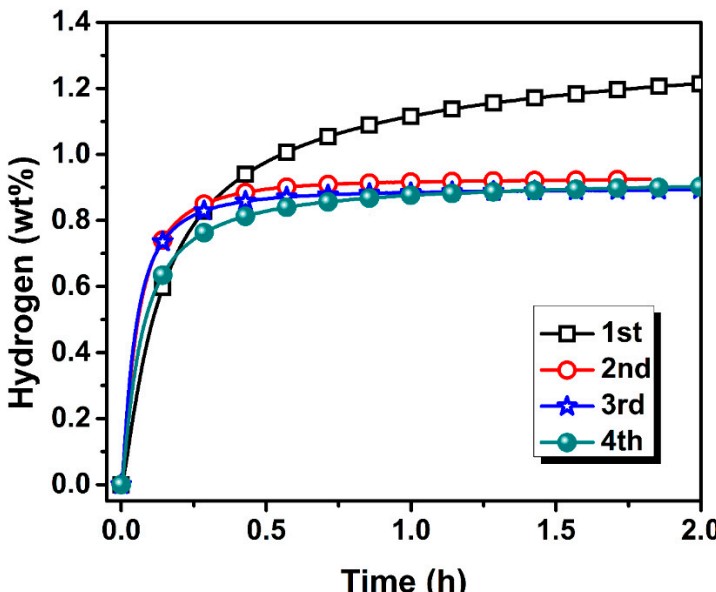

**Figure 8.** Hydrogenation kinetics of 30 min BM TiFe + 1 wt.% KCl sample at 298 K under 20 bar of hydrogen pressure.

To further investigate the effect of other salts, especially chloride salts, a set of samples were prepared by 30 min ball milling of TiFe with 1 wt.% of sodium chloride (NaCl), calcium chloride ($CaCl_2$), ferric chloride hexahydrate ($FeCl_3 \cdot 6H_2O$) and potassium iodide (KI), separately. The hydrogenation kinetics of these TiFe with salt addition are shown in Figure 9. The effect of KCl addition is shown in the figure for comparison. It can be seen from the figure that after the addition of 1 wt.% NaCl, $CaCl_2$ and KI salts, hydrogenation kinetics are similar and around 1.2–1.3 wt.% of hydrogen was absorbed by these samples after 10 h of hydrogenation. However, in case of the hydrated salt ($FeCl_3 \cdot 6H_2O$), slow hydrogenation kinetics could be seen. This sample absorbed only around 0.4 wt.% of hydrogen after 10 h and took more than 30 h to reach full capacity. The surface of TiFe may be oxidized in presence of the hydrated salts during the ball milling process. Thus, it shows slower first hydrogenation kinetics than TiFe with the other salts.

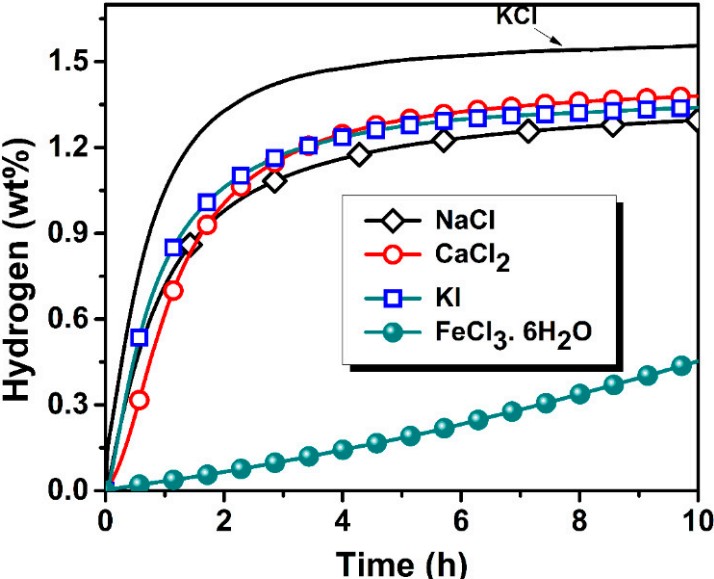

**Figure 9.** Effect of other salts (1 wt.% of NaCl, $CaCl_2$, KI and $FeCl_3 \cdot 6H_2O$) on the first hydrogen kinetics of 30 min BM TiFe at 298 K under 20 bars of hydrogen pressure.

## 4. Conclusions

The effect of adding salts to TiFe by ball milling has been investigated. We found that ball milling of TiFe with KCl salt improved the first hydrogenation kinetics. Ball milling TiFe with KCl reduces the crystallite and particle sizes. The synthesized samples show chemical homogeneity after ball milling. The TiFe milled with KCl shows similar first hydrogenation kinetics as ball-milled pure TiFe. However, hydrogen storage capacity of TiFe + 1 wt.% KCl is around 1.5 wt.%, which is much higher than the pure TiFe (1.1 wt.%). It was observed that, with an increase in KCl amount and BM time, first hydrogenation kinetics are improved. However, in both cases, storage capacity decreased. Other salts (NaCl, $CaCl_2$, KI) also improved the first hydrogenation kinetics but the hydrogen capacity was reduced. Milling of TiFe with hydrated iron chloride salt ($FeCl_3 \cdot 6H_2O$) resulted in a slow kinetics. In conclusion, milling with salts was shown to be beneficial because the hydrogen capacity was higher than when milling was performed on pure TiFe.

**Author Contributions:** Conceptualization, J.H. and J.M.; methodology, J.H. and J.M.; data curation and investigation, J.M.; writing—original draft preparation, J.M.; writing—review and editing, J.H. and J.M.; supervision, J.H. All authors have read and agreed to the published version of the manuscript.

**Funding:** This research received no external funding.

**Data Availability Statement:** Not applicable.

**Acknowledgments:** J.M. is thankful to Fonds de recherche Nature et technologies Québec (FRQNT) for postdoctoral scholarship.

**Conflicts of Interest:** The authors declare no conflict of interest.

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
