# Peer review of "Effect of KCl Addition on First Hydrogenation Kinetics of TiFe"

_compounds, doi:10.3390/compounds2040020_

Round 1

Reviewer 1 Report

The research presented by Joydev Manna and Jacques Huot it's interesting. The author employed valuable techniques allowing for most of the given conclusions. The manuscript is scientifically sound, and I am expecting high interest of the readership. However, some points must be improved and clarified, before the article can be considered for publication.

- It is unclear why this work is interesting. That is, work motivation is absent in the introduction.

- The introduction of relevant background and research progress was not comprehensive enough

The authors do not justify the potassium chloride (KCl) salt concentrations used in this system. Due to the importance for this system, a reasoned explanation must be attached as part of the motivation and justification of the work.

- The quality of all figures should be improved

- Information on lattice strain must be provided.

- Miller indices for some of the peaks in figure 1b were not identified. Please identify

-Crystallographic cards for the XRD patterns must be provided.

- The discussion of XRD patterns and structural parameters should be enriched with a stronger discussion, comparing the results obtained with others reported in the literature and strongly discussing the changes obtained from a physical point of view.

- The description of images and size distribution should be improved, depending on the changes and the mechanisms that influence the observed variations. In its present form, the text is very qualitative and descriptive.

- It is appreciable that the authors have analyzed the Effect of KCl Addition on First Hydrogenation Kinetics of TiFe, but there is no in-depth scientific discussion. A careful review of the entire manuscript must be performed. Strong scientific discussions must be made throughout the entire manuscript.

- More current references should be included to highlight the timeliness of the research

In summary, a large part of the discussion and interpretation of the experimental data is not scientifically justified.

Author Response

We would like to thanks the reviewer for his comments. Below are the specific answers.

- It is unclear why this work is interesting. That is, work motivation is absent in the introduction.

Ans: Thank you reviewer for the valuable comment. We have included the motivation of the work in the introduction part of the paper.

- The introduction of relevant background and research progress was not comprehensive enough

Ans: A paragraph on current research progress has been added in the introduction part of the manuscript.

The authors do not justify the potassium chloride (KCl) salt concentrations used in this system. Due to the importance for this system, a reasoned explanation must be attached as part of the motivation and justification of the work.

Ans: The used concentration of KCl is 1 wt% and 5 wt%. We have added it now in the experimental section.

- The quality of all figures should be improved

Ans. High quality images with desired quality (>300 dpi) are included in the manuscript.

- Information on lattice strain must be provided.

Ans. Lattice strain for TiFe, TiFe after KCl addition are added in the manuscript.

- Miller indices for some of the peaks in figure 1b were not identified. Please identify

Ans: Miller indices are included in the Figure 1b.

-Crystallographic cards for the XRD patterns must be provided.

Ans. JCPDS card number for XRD patterns are provided

- The discussion of XRD patterns and structural parameters should be enriched with a stronger discussion, comparing the results obtained with others reported in the literature and strongly discussing the changes obtained from a physical point of view.

Ans. A discussion is added with the results and discussion part of the x-ray diffraction analysis and compared with the reported literature.

- The description of images and size distribution should be improved, depending on the changes and the mechanisms that influence the observed variations. In its present form, the text is very qualitative and descriptive.

Ans. Description of SEM figures and size distribution are changed, and comparative part is included to show the similarity with the reported literature.

- It is appreciable that the authors have analyzed the Effect of KCl Addition on First Hydrogenation Kinetics of TiFe, but there is no in-depth scientific discussion. A careful review of the entire manuscript must be performed. Strong scientific discussions must be made throughout the entire manuscript.

Ans. A discussion has been added in the results and discussion section.

- More current references should be included to highlight the timeliness of the research

 Ans. Current references are included.

In summary, a large part of the discussion and interpretation of the experimental data is not scientifically justified.

Ans.: We added some text to the manuscript to address this remark.

Reviewer 2 Report

This paper stated the additives of KCl on the activation of TiFe alloy. The reason of the motivation of this work should be addressed in the introduction parts. The advantage of this additives should be clearly state. The machnism of the activation process by the addition of KCl should be explained with more supportive data like XPS, etc. The recent new literature reference in 2021 and 2022  should be cited. After these modifications, this paper can be accepted. 

Author Response

We thank the reviewer for this valuable comment. As per the suggestion, motivation behind the work has been incorporated in the introduction section of the manuscript. Recently reported literature has been included with the manuscript.

Reviewer 3 Report

In this paper, the effect of the addition of potassium chloride (KCl) by ball milling on the 9 first hydrogenation kinetics of TiFe is reported. The research results are convincing and meaningful. It is recommended for publication after some revisions.

1.     The information on other salts (NaCl, CaCl2, KI, etc.) used in this work should be included in the experimental section.

2.     The scale bar of Figure 4 (a) is difficult to identify.

3.     The authors should clarify the methods of the determination of lattice parameters and crystallite size in this paper. Are they obtained by the Rietveld refinement? What is the Rwp/Rp and the goodness of fit?

4.     The authors found that the as-cast TiFe ball milled with KCl addition exhibited higher hydrogen storage capacity than the as-milled TiFe, but they did not provide any explanations. The as-milled TiFe of this work only shows ~1.1 wt% of hydrogen capacity, which is evidently lower than the reported results. For example, Emami et al (Emami, Hoda, et al. Acta Materialia 88 (2015): 190-195) reported that the as-milled TiFe has a hydrogen capacity of 1.5wt%. The authors should compare with other works and discuss the hydrogen capacities of the ball-milled TiFe with or without KCl addition.

5.     The authors claim that the total storage capacity of 5% KCl added TiFe is found to be around 1.34 wt.% which is less than the 1 wt.% KCl added sample. They said, “The decrease in hydrogen storage capacity for higher salt amount is not clear and may need further investigation.” The KCl could not react with hydrogen, lower capacity should be expected on the alloy with higher KCl addition.

Author Response

We thanks the referee for his valuable comments. Below are the answer to the specific questions.

  1. The information on other salts (NaCl, CaCl2, KI, etc.) used in this work should be included in the experimental section.

Ans: Information on other salts are added on the experimental section.

  1. The scale bar of Figure 4 (a) is difficult to identify.

Ans: Scale bar is re-written with bigger and bold font.

  1. The authors should clarify the methods of the determination of lattice parameters and crystallite size in this paper. Are they obtained by the Rietveld refinement? What is the Rwp/Rp and the goodness of fit?

Ans. Yes, lattice parameters and crystallite size were determined using Rietveld refinement and now mentioned in the result and discussion section. The Rwp, Rp values are mentioned in Fig 1d. Rwp/Rp is around 1.25 and goodness of fit is around 1.2.

  1. The authors found that the as-cast TiFe ball milled with KCl addition exhibited higher hydrogen storage capacity than the as-milled TiFe, but they did not provide any explanations. The as-milled TiFe of this work only shows ~1.1 wt% of hydrogen capacity, which is evidently lower than the reported results. For example, Emami et al (Emami, Hoda, et al. Acta Materialia 88 (2015): 190-195) reported that the as-milled TiFe has a hydrogen capacity of 1.5wt%. The authors should compare with other works and discuss the hydrogen capacities of the ball-milled TiFe with or without KCl addition.

Ans: A paragraph and a table are added in the results and discussion section of the manuscript to compare the obtained results with the reported activation processes and their effect on first hydrogenation kinetics of TiFe. It has been noted that the experimental condition used in the paper of Emami et al (100 bar H2 pressure, 30°C temperature) and this manuscript (20 bar hydrogen pressure, 25°C) are different. This would be the plausible reason for the difference in absorbed hydrogen amount.

  1. The authors claim that the total storage capacity of 5% KCl added TiFe is found to be around 1.34 wt.% which is less than the 1 wt.% KCl added sample. They said, “The decrease in hydrogen storage capacity for higher salt amount is not clear and may need further investigation.” The KCl could not react with hydrogen, lower capacity should be expected on the alloy with higher KCl addition.

Ans. From literature, the capacity of TiFe is 1.86 wt.%. We agree that KCl does not absorb hydrogen thus, adding 1wt.% and 5wt.% of KCl will reduce the maximum capacity to respectively 1.84 wt% and 1.77 wt.%. Therefore, the higher decreases of capacity for the 5wt.% KCl could not be attributed only to the ‘dead weight’ of the KCl. This is now discussed in the manuscript.

Round 2

Reviewer 1 Report

Authors carefully review the manuscript, improving the quality. Considering this, I recommend the work for publication in this prestigious journal